# A Qualitative Study of Health-Related Experiences Associated with Lifestyle Role Transitions Among Local Residents in Their 60s

**DOI:** 10.3390/healthcare13212702

**Published:** 2025-10-26

**Authors:** Hiroko Nakano, Mikako Arakida

**Affiliations:** 1Department of Nursing, University of Occupational and Environmental Health, Kitakyushu 807-8555, Japan; 2Department of Nursing, Kawasaki City College of Nursing, Kawasaki 212-0054, Japan

**Keywords:** role, adaptation, psychological, health status, quality of life, aged

## Abstract

**Background/Objectives:** As population aging garners attention worldwide, there is great significance in communicating information on such measures to countries outside of Japan, which is considered unique in its position as a “super-aging society.” This study objectives to investigate public health measures linked to daily life by clarifying how the role transitions of local residents in their 60s, such as seeking re-employment, looking after grandchildren, and caring for family, affect their health status. **Methods:** We conducted focus group interviews with 26 residents and analyzed them qualitatively and inductively. **Result:** The findings suggested that, in predicted role transitions voluntarily chosen by participants, they tended to experience positive changes in health through the transition, although temporary feelings of fatigue were also described in relation to re-employment and grandchild care. Even in anticipated role changes, some participants expressed reluctance to engage in health-promoting activities within the local community. In cases of unavoidable role transition to family caregiving, participants described difficulties in maintaining self-care and feelings of caregiving fatigue that were challenging to manage through personal effort alone. These findings suggest that health support during role transitions in one’s 60s may benefit from including information about community activities and opportunities to build connections with local residents. In addition, support for those transitioning into caregiving roles could focus on facilitating access to social resources and tailoring assistance to individual needs. **Conclusions:** This study confirmed to specifically target health support for people in their 60s based on the results of this study, the focus on the transition needs to include not only role transition to re-employment but also unavoidable transition to caregiving.

## 1. Introduction

As the increasing elderly population is becoming a worldwide topic of debate, the United Nations (UN) has unveiled the “UN Decade of Healthy Ageing” (from 2021 to 2030) implemented by the World Health Organization (WHO) to outline issues and countermeasures for the increasing population of elderly people aged 60 years and over [1]. The aging rate in Japan is particularly high when considered globally, which makes preserving the mental and physical functions of people aged 75 years and older a pressing issue from the viewpoint of the sustainability of the country’s social security system. Japan’s Ministry of Health, Labour and Welfare (MHLW) is aiming to maintain health in elderly people by preserving their social roles, with one of the targets established in “Health Japan 21 (the second term)” increasing the number of elderly people who engage in social activities [2].

In terms of social roles, as over 90% of companies have a mandatory retirement-age system and the mandatory age of retirement is 60 years at over 70% of companies, many people in Japan reach such age in their early 60s [3]. In recent years, the elderly’s employment rate has been increasing as more people are utilizing re-employment systems to return to their jobs after mandatory retirement [3]. Looking at household roles of people aged in their 60s, the percentage of people in Japan who are the primary caregiver for a recipient of long-term care with whom they live is approximately 30% [4] for people aged 60–69 years, which is higher than for any other age range. The average age at which their first grandchild is born is also in their 60s [5]. In Japan, returning to their parent’s home to give birth is customary for women. In recent years, the “*sofubo techou* (grandparent handbook)” has been utilized at multiple local municipalities, clarifying the social needs for grandparents to be involved in raising grandchildren.

Reports from overseas indicate that the proportion of grandparents providing childcare varies significantly by country (24% to 60%), and that the stability of grandparent childcare is high and trending upward [6]. Regarding re-employment, at least 26% of those who initially left the workforce later returned to employment. Using a different definition of employment (“hours-only” definition), this figure is reported to be approximately 39.7% [7]. For caregiving, among study participants who were not caregivers at baseline, 11% reported becoming caregivers approximately 12 years later [8]. While these results do not yield a consistent frequency, it is clear that a certain number of individuals experience role transitions during later life. Therefore, we decided to focus on people in their 60s, who are likely to experience role transitions about re-employment, looking after grandchildren, and caring for family.

Meleis’ transition theory classifies individual transition types into developmental, situational, and health/illness categories [9]. According to Meleis’ transitions theory, caregiving and re-employment are categorized as situational transitions, while becoming a grandparent is classified as a developmental transition. Role transitions are considered related to the cycle of human health and illness [9]. Regarding re-employment, based on Atchley’s Continuity Theory, continuing work or re-employment can serve as a means to maintain past identities and life rhythms, preserving self-efficacy [10]. However, re-employment involves adapting to new environments and changing role expectations, making psychological adaptation crucial. A scoping review on physical and mental function after retirement in Western countries indicates that post-retirement function may change in various ways, including decline or improvement [11]. Therefore, implementing preventive health measures for people approaching retirement is considered crucial for maintaining subsequent health [11].

Caring for a spouse, parent, or other close relative often begins suddenly and unexpectedly, significantly altering previous lifestyle habits and self-identity. Meleis’ transitions theory reports that such “unexpected transitions” are prone to involve confusion and stress reactions [12]. Furthermore, the transition into caregiving has been reported to be associated with a significant increase in perceived stress and depressive symptoms, as well as a decline in health-related quality of life, following the onset of caregiving [8,13]. Previous research indicates that grandparents caring for their grandchildren positively impacts the grandparents’ health [14]. On the other hand, prior research shows that the role of grandparents can be perceived as either fulfilling or burdensome, depending on the level of support provided and the circumstances involved [15].

Medical statistics in Japan show that the age of 60 onward is characterized by susceptibility to lifestyle-related diseases, with increased disease prevalence rates and ratio of people with subjective symptoms [4]. Additionally, many life events that could affect health also happen at this age [16]. In Japan, the long-term care insurance system was established in 2000, enabling people aged 65 years or older requiring support or long-term care regardless of the reason to receive such long-term care services for a copayment of 10% or 20%. The continuing increase in the number of people certified as needing long-term care and who can therefore receive long-term care benefits is an issue currently faced by society [3]. Therefore, the type and timing of intervention for people in their 60s, who are on the cusp of becoming elderly, needs to be discussed in detail based on the results of investigation of healthcare measures for the population of people aged 75 years or older—a segment that is expected to increase rapidly going forward.

Therefore, this study aims to investigate public health measures linked to daily life by clarifying how the role transitions of local residents in their 60s, such as seeking re-employment, looking after grandchildren, and caring for family, affect their health status. Previous studies have reported that in health support for older adults, it is necessary to adopt an approach that focuses not only on the health benefits of activities, but also on the diverse personal and important goals of older adults themselves [17]. Health-support measures being offered to people in their 60s in Japan who experience elderly role transition fend off declines in mental and physical function, even if slightly. As population aging garners attention worldwide, there is great significance in communicating information on such measures to countries outside of Japan, which is considered unique in its position as a “super-aging society.”

## 2. Materials and Methods

### 2.1. Target Areas (2020 National Census)

Town A, the subject of this study, has a low percentage of the population aged 65 and over 17.9% (31.5% nationally) and a low percentage of three-generation households 3.4% (4.2% nationally), and it is located in the suburbs of a large city in Japan. Town A was originally a thriving agricultural community but has recently developed into a bedroom community, and its population has increased rapidly. Town A has a population of about 48,000.

### 2.2. Participants in the Study

We asked municipal officers in Town A’s division for older adults’ affairs to identify residents of Town A in their 60s who had experienced a change in their life role, either in re-employment or in caring for grandchildren or others.

### 2.3. Survey Method

Transition theory states that an individual’s response to transition is related to their state of health and that paying attention to the personal events of each individual’s life is essential [6]. In this study, we decided to use qualitative research methods to gain a deeper understanding of the issues of motivation and functional changes in role transition after retirement. We also used the focus-group interview method to investigate the experiences of people in their 60s and elicit discussion through mutual interaction between participants.

### 2.4. Data Collection and Analysis Methods

Recruitment for this study’s participants was conducted by staff members of Town A. Group interviews were conducted with 4 to 6 participants and lasted 70 to 90 min. The interviews were held indoors at a welfare center where privacy could be ensured. The researcher created an interview guide based on the study objectives and obtained confirmation from the co-researcher. The interview guide consisted of sections on life changes experienced in one’s 60s, physical condition, health promotion activities, and perceptions of fatigue. The first author, who has experience as a public health nurse, interviewed each group (Appendix A). The first author practiced role-playing and mock interviews within the research team prior to the interviews, striving to ensure reliability and validity. Groups were organized based on the schedule. As a basic principle, each group consisted of participants of the same gender, but two groups included one male participant among the women. Participants’ residential areas were not considered.

The analysis procedure involved transcribing the recorded interviews verbatim and conducting qualitative descriptive analysis. The verbatim transcripts were carefully read, and participant statements expressing meaning relevant to the research objectives were extracted and assigned codes representing their content. The initial coding for this study was conducted by two individuals: the first author and a co-researcher. First, the first author carefully read the verbatim transcripts, extracted meaning relevant to the research objectives, and coded it. Subsequently, the co-researcher similarly reviewed the content, and both repeatedly discussed the validity of codes and categories. In the initial phase, they independently reviewed the verbatim transcripts and later reached consensus through discussion. This study, grounded in qualitative descriptive research, aimed to describe and organize participants’ narratives as faithfully as possible. Quantitative reliability measures (e.g., Cohen’s kappa) were not calculated, as such metrics were deemed unsuitable for analyses emphasizing contextual and semantic interpretation, such as this study. Instead, the validity and reliability of the analysis were ensured through independent verification by multiple researchers and ongoing discussion.

Similar codes were then grouped to form subcategories. Subsequently, categories were extracted by comparing similarities and differences while increasing the level of abstraction. The methodological foundation of this study was strictly qualitative descriptive analysis, aiming to describe and organize experiences in everyday language as faithfully as possible to the participants’ narratives. During the stage of examining relationships between categories, Glaser’s “theoretical coding family” was referenced as an auxiliary framework. This was employed not for theory generation, but as a conceptual guide to structure causal relationships and connections between categories. This enabled the organization of relationships such as “Can this category be the cause of another?” and the creation of relationship diagrams.

All interviews were conducted in Japanese (the native language of both the researcher and participants). The interview excerpts presented in this paper were translated from the original Japanese transcripts into English. To enhance the validity of the analysis, we repeatedly analyzed and discussed the data with input from several experts in public health nursing. We also regularly reviewed the categories and verbatim transcripts among the researchers until consensus was reached. Saturation was determined to have been reached after the fourth group, as few new codes or themes emerged and the main categories were repeatedly confirmed. CAQDAS (NVivo 1.5) was used for the analysis. The researchers reported the research methods, study design, analysis, and results (incorporating the COREQ framework) according to the Consolidated Standards of Reporting Qualitative Research (COREQ) checklist [18].

### 2.5. Ethical Considerations

We explained to the study participants that participation in the study was voluntary, that they would not be disadvantaged by nonparticipation, that anonymity was guaranteed, and that the data would be kept strictly confidential. We obtained written consent from them. All data obtained were analyzed after anonymization. This study was approved by the Institutional Review Boards of University of Occupational and Environmental Health, Japan (Approval No.18-Ifh-034) and of International University of Health and Welfare University (Reception No.H30-200).

## 3. Results

### 3.1. Background of Participants in the Study (Table 1)

We used opportunity sampling and asked 26 participants to participate in the interviews; all 26 agreed to cooperate. The participants were 26 Town A residents in their 60s (7 men and 19 women). They were taking life roles in re-employment (n = 17), looking after grandchildren (n = 10), and caring for family members (n = 17). Fourteen had more than one role. Retirement was the most common experience cited as a transition in life role (n = 12). The participants were divided into six groups for the interview.

### 3.2. Interview Results (Table 2)

After analyzing the interviews of the six groups, we extracted 20 categories and 52 subcategories, 110 codes.

The categories are explained below according to their domain. The domains, categories, and subcategories are shown in Table 2. The following sections explain the categories listed above and the narratives of participants representing each category, with IDs added to the end of each.

Domain: BoldCategory: ItalicSubcategory: Double quotation

### 3.3. Process of Role Transition

#### 3.3.1. Role Transitions That Were Predicted and Chosen by the Participant

Many participants who experienced the transition from full-time employment to mandatory retirement were able to anticipate their retirement in advance and had “gathered information for their post-retirement lifestyle.” Other participants, after “choosing retirement based on their own will,” pursued “re-employment for personal motivations” such as wanting to be useful to the society. Others accepted roles such as “looking after grandchild/ren, as expected” or “providing caregiving for parent/s, as expected.”


*“I retired at 65 years of age. About a month ago, I started looking for outdoor places on the Internet because I was afraid that I’d end up staying indoors if I failed to find them.”*
(5)


*“Caring for my mother-in-law also started naturally. It’s not like I’m doing it because I have to, but naturally. We’ve been together for over 40 years, so after I retired, I sort of started looking after my parents. It just happened naturally.”*
(12)

**Table 1 healthcare-13-02702-t001:** Participants’ demographic profile.

				Family Composition			Pre-Transition Roles			A Change in Their Life Role					
Gp	Id	Age	Gender	Married Couple	Married Couple and Parents	Other	Working	Occupation	House Wife	Re Employment	Occupation	Caring for Grandchildren	Residence	Family Care Giving	Care Level
A	1	67	M	√			√	Civil servant				√	Nearby		
	2	64	M	√			√	Civil servant		√	Agriculture			√	Mild
	3	67	M	√			√	Company employee		√	SC	√	Nearby	√	Mild
	4	66	M			Living Alone	√	Healthcare professional		√	SC				
	5	68	M	√			√	Company employee		√	SC				
B	6	69	F	√					√	√	SC	√	Nearby	√	Mild
	7	62	F	√	Parents				√					√	Mild.Severe
	8	68	F	√					√						
	9	65	F	√	Parent		√	Educator		√	Educator			√	Mild
	10	66	F	√					√	√	Agriculture	√	Co-Nearby		
	11	67	F		Parents		√	Company employee		√	SC				
C	12	66	F	√	Parents		√	Company employee		√	SC	√	Nearby	√	
	13	63	F	√			√	Company employee		√	SC	√	Nearby	√	
	14	65	F	√			√	Company employee				√	Distant	√	
	15	69	M	√			√	Company employee		√	SC				
D	16	63	F		Parents		√	Company employee						√	Moderate
	17	65	F	√			√	Civil servant		√	Nursing Care			√	
	18	67	F	√					√			√	Nearby	√	Mild
	19	68	F	√					√	√	SC	√	Nearby		
E	20	63	F	√	Parent		√			√	Nursing Care			√	Moderate
	21	65	F	√	Parent		√							√	Moderate
	22	61	F	√					√					√	Severe
F	23	67	F	√			√	Company employee		√	SC				
	24	69	M	√	Parents		√	Self-employed			Self-employed			√	Moderate
	25	66	F	√		Child			√			√	Nearby	√	Severe
	26	60	F	√			√	Healthcare professional		√	Healthcare professional			√	Mild

SC: Silver Human Resources Center.

**Table 2 healthcare-13-02702-t002:** Category structure.

Major Categories	Categories	Sub Categories
Process of role transition	Role transitions that were predicted and chosen by the participant	Gathered information for their post-retirement lifestyle
		Choosing retirement based on their own will
		Re-employment for personal motivations
		Looking after grandchild/ren, as expected
		Providing caregiving for parent/s, as expected
	Inevitable role transition to family caregiving	Family caregiving when faced with the sudden illness of a spouse
		Starting caregiving for bereaved parent after losing other parent owing to death
Reactions to role transitions	Release from stressors	Released from the stress of work after retirement,
		Ending relationship with neighborhood upon retirement
		Release from the struggle between work and family caregiving.
	Sense of fulfillment from new role	Feeling a different type of job satisfaction with re-employment
		Sensing a loving bond for grandchild through caring for him/her
		Providing caregiving with a sense of gratitude and self-development
	Sense of having extra time	Adjust their own schedule
		Peace of mind after retirement
	Confusion and sense of isolation felt after retirement	Confusion on how to spend the extra time after retirement
		Felt isolated after retirement
	Confusion at the role transition to family caregiving	Psychologically distressed owing to the caregiving that they were forced to provide for their parent/s
		Confusion regarding family caregiving that was suddenly necessary
	Caregiving lifestyle with no leeway	Daily lifestyle in which family caregiving makes relaxation impossible
		Difficulty leaving the house freely because of caregiving for my parent
Effects on self-care and health status	Self-care identified from experiences	The effects of exercise identified based on experience
		Things that are important for health maintenance
	Self-care made difficult by family caregiving	I prioritize family caregiving ahead of my own medical examinations
		I stopped exercising when I started providing family caregiving
	Positive changes in health awareness and health behaviors associated with role transitions	It is better than when I was really busy during employment
		Re-employment helps me to maintain my health
		I started exercising after retirement
	Temporary fatigue associated with re-employment	My lower back hurts when I work
		It is confusing because my lifestyle rhythm does not match my re-employment
	Temporary sense of fatigue after looking after grandchildren	I feel a sense of physical and mental fatigue from looking after my grandchild/ren
		I can’t help but try too hard to look after my grandchild/ren
		I feel pushed to my limit in caring for my grandchild/ren for long periods of time
	Reluctance to participate in health promoting activities in local communities	I don’t feel connected to my local community and participating in workshops is bothersome
		I feel reluctant to engage in group exercise
	Caregiving fatigue that is difficult to cope with by personal effort alone	Chronic insomnia
		Psychological conflict accompanying family caregiving
		Joint pain associated with physical caregiving
Needs to maintain health	Motivation to maintain my health	Motivation to maintain my health
		Hoping to start exercising
		I have hopes for facility usage discounts for when I am 60 years or older.
		Continuing methods to maintain health based on experience
	Hopes to connect with local communities	I am making friends in the community by volunteering
		I want there to be activities for men
	Utilization of social resources for caregiving	I feel relief when the care recipient is out of the house to receive services
		I sleep deeply when the care recipient is away overnight
	Self-care for a lifestyle involving family caregiving	Making efforts to adjust his/her mood despite providing caregiving daily
		Alleviating stress through exchanges with visiting nursing care staff
		Let go of the stress of caregiving through exchanges with friends
		Exercise in between family caregiving

#### 3.3.2. Inevitable Role Transition to Family Caregiving

Study participants who accepted a family caregiving role underwent this as an unavoidable role transition, with comments such as, “family caregiving when faced with the sudden illness of a spouse” or “starting caregiving for bereaved parent after losing other parent owing to death.”


*“I worked for many years, but after my father passed away, my mother was left alone. As it became difficult for her to live on her own, I had no choice but to quit my job. Now I live with my husband and mother, and I look after her.”*
(12)

### 3.4. Reactions to Role Transitions

#### 3.4.1. Release from Stressors

Participants experienced release from stressors up to that point as a result of retiring from work, with comments such as “released from the stress of work after retirement,” ending relationship with neighborhood upon retirement” and “release from the struggle between work and family caregiving.”


*“After I left the company, I was under a lot of mental stress, and now I think that it was very hard for me at that time. Mentally I feel a lot better now because I’m free from that.”*
(5)


*“I quit my neighborhood association after I retired. When I was still working, I joined and put up with it because I had to socialize. But after I retired, I thought I’d had enough of it and quit the association, and the stress went away.”*
(1)

#### 3.4.2. Sense of Fulfillment from New Role

Participants felt a sense of fulfillment from their new role, with comments such as, “feeling a different type of job satisfaction with re-employment,” “sensing a loving bond for grandchild through caring for him/her” and “providing caregiving with a sense of gratitude and self-development.”


*“I have come to realize how enjoyable it is to work, not for money or anything like that, but for the vitality of my life, for the feeling that I can be useful, and for my sense of fulfillment… I now think that working is enjoyable for the first time.”*
(26)


*“Last year, I had my first grandchild, and looking after my grandchild from time to time is now my source of energy.”*
(21)

#### 3.4.3. Sense of Having Extra Time

Participants could “adjust their own schedule” and felt “peace of mind after retirement.”


*“When I retired from work, I didn’t have to get up so early even when I was looking after my grandchildren, so I had more time to spare. Now I’m looking after my grandchildren with a relaxed mind…”*
(13)

#### 3.4.4. Confusion and Sense of Isolation Felt After Retirement

Participants who experienced retirement felt “confusion on how to spend the extra time after retirement.” Unmarried participants “felt isolated after retirement” when they stopped interacting with their workmates owing to retirement. These were both comments from male participants:


*“Since I didn’t work, I had nothing to do but wander around the park and go fishing.”*
(15)


*“I’m a single person, so I’ve felt that sense of loneliness more since I retired. I don’t have any grandchildren or a partner.”*
(4)

#### 3.4.5. Confusion at the Role Transition to Family Caregiving

Participants felt “psychologically distressed owing to the caregiving that they were forced to provide for their parent/s,” with comments including “confusion regarding the family caregiving that was suddenly necessary.”


*“I loved my job, so I never thought I would quit it for providing caregiving… It was a job that I could keep for a long time. So quitting it was very stressful…”*
(21)

#### 3.4.6. Caregiving Lifestyle with No Leeway

Participants who provided family caregiving experienced a “daily lifestyle in which family caregiving makes relaxation impossible,” always being concerned about the time and sometimes feeling as if they were being called even though they were not being called. Participants who provided family caregiving described “difficulty leaving the house freely because of caregiving for my parent.”


*“At about 3 p.m., I think, Oh, I have to go home now. Even when I went out, I would say, “Oh, I have to go home soon.” That is, I don’t really have much free time.”*
(7)


*“When there is an event, we will coordinate the nursing care services in advance. When the coordination is complete, I can go out for a day of training, etc.”*
(7)

### 3.5. Effects on Self-Care and Health Status

#### 3.5.1. Self-Care Identified from Experiences

Participants spoke about “the effects of exercise identified based on experience” as they had personally experienced this, and “things that are important for health maintenance,” such as stress management and sleep adjustment.


*“I’m really glad I found that exercise. Basically, I’m feeling better now.”*
(10)


*“I really need to relieve some stress. From my experience, I think that the most important thing for health management is sleep.”*
(20)

#### 3.5.2. Self-Care Made Difficult by Family Caregiving

Caregivers found engaging in self-care difficult, experiencing things such as being late to medical examinations owing to prioritizing their husband’s caregiving, with comments such as, “I prioritize family caregiving ahead of my own medical examinations” and “I stopped exercising when I started providing family caregiving.”


*“My husband has a heart condition and other illnesses, so I go to the hospital for him. But I can’t easily go to the hospital for myself.”*
(1)

#### 3.5.3. Positive Changes in Health Awareness and Health Behaviors Associated with Role Transitions

Participants felt that they currently had better health awareness and health behaviors than prior to retirement, with comments such as “it is better than when I was really busy during employment,” “re-employment helps me to maintain my health,” and “I started exercising after retirement.”


*“Looking back on it now that I’ve quit the company, the mental stress was pretty tough for me too. It was a real burden. At the worst times, I couldn’t sleep… Even though there were times like that, I felt a lot more relaxed mentally after I quit.”*
(5)


*“They will be happy if I go to work as a caregiver. If I do something for them, they will be pleased, so it’s more of a pleasure and I don’t feel tired.”*
(17)


*“After I retired, I started going to the training room and I also run in the park. I hope that I can exercise and somehow extend my healthy life expectancy and pass away.”*
(3)

#### 3.5.4. Temporary Fatigue Associated with Re-Employment

Participants made comments such as “my lower back hurts when I work” and “it is confusing because my lifestyle rhythm does not match my re-employment.”


*“I sweep twice a week as part of my work. That’s hard work now. My back hurts when I move a bit.”*
(15)


*“I worked at a fish market for 40 years, so now the hardest thing is that I can’t help waking up early in the morning. Even if I watch TV until midnight or 1am, I still wake up at around 4 a.m. I can’t get out of that rhythm, so it’s tough.”*
(15)

#### 3.5.5. Temporary Sense of Fatigue After Looking After Grandchildren

Participants who looked after grandchildren made comments such as “I feel a sense of physical and mental fatigue from looking after my grandchild/ren,” “I can’t help but try too hard to look after my grandchild/ren,” and “I feel pushed to my limit in caring for my grandchild/ren for long periods of time.”


*“Even if I think, “Oh, today is a bit tough,” I feel energized when my grandchildren come and play with me. I feel a bit excited and energetic, even though I have to pay attention to my grandchildren. Honestly speaking, I feel tired after they leave. It doesn’t mean that I am compelled to go to bed, though.”*
(14)


*“When my grandchild’s parents come to pick him up, my honest feeling is “Go home quickly.”*
(12)

#### 3.5.6. Reluctance to Participate in Health Promoting Activities in Local Communities

Participants made comments such as “I don’t feel connected to my local community and participating in workshops is bothersome” and “I feel reluctant to engage in group exercise.”


*“I’m often told to exercise, and my husband asks me to go to the pool or go for a walk, but for some reason, I’m not very good at it.”*
(21)


*“Even if it’s a light dance, I can’t do it because it’s not for me. I prefer exercise that I can do at my own pace.”*
(25)


*“I’ve been working all the time, so I don’t have any connections in the community. I find taking the first step hard.”*
(16)

#### 3.5.7. Caregiving Fatigue That Is Difficult to Cope with by Personal Effort Alone

Some participants providing caregiving exhibited “chronic insomnia.” Participants providing family caregiving also had mixed feelings about caregiving and experienced “psychological conflict accompanying family caregiving” owing to fatigue from interpersonal relationships within their household as a result of caregiving. As a result, they noticed “joint pain associated with physical caregiving.”


*“I’m sleeping in another room, but I feel like I’m being called even when I’m not. So I haven’t had a good night’s sleep lately.”*
(7)


*“I can’t take care of him if I’m too tired, and if I’m too tired, I don’t even want to see his face. But I can’t miss giving him his meals, so there is a conflict.”*
(22)

### 3.6. Needs to Maintain Health

#### 3.6.1. Hopes to Maintain Health

Participants focused on maintaining their health as they underwent role transition. Participants described positive feelings, with comments such as, “motivation to maintain my health,” “hoping to start exercising,” and “continuing methods to maintain health based on experience.” Regarding a discount service at a municipal facility for persons aged 65 years or older in Town A, participants described hopes for the enhancement of specific services, with one commenting, “I have hopes for facility usage discounts for when I am 60 years or older.”


*“If I were to participate in the walking event, I would receive points if I were over 65 years old. However, if I were over 60 years old, I would receive nothing even if I went. I want people to start from the age of 60.”*
(16)


*“My goal is to maintain a regular lifestyle and physical strength until I’m 70 years old.”*
(26)


*“I want to be able to do the training and exercise I’m doing now even when I’m in my 70s or 80s.”*
(3)

#### 3.6.2. Hopes to Connect with Local Communities

Role transition encouraged participants to change their viewpoint to focus on activities in the local community, with comments such as “I am making friends in the community by volunteering.” A male participant described participating in a cooking workshop for men, saying, “I want there to be activities for men.”


*“I want to find volunteer activities in the area where I live and try them out. I want to use them to build connections in the community.”*
(23)


*“At the cooking class for men, the men just kept talking and talking, and they just wouldn’t stop. It was like, “No, you have to cook!” I wish there was a place like that.”*
(5)

#### 3.6.3. Utilization of Social Resources for Caregiving

Participants providing caregiving made comments such as “I feel relief when the care recipient is out of the house to receive services” and “I sleep deeply when the care recipient is away overnight.” These comments demonstrated the need for people providing caregiving to utilize social resources to secure time to rest as well as free time.


*“This year, my husband went on a short stay for the first time, staying overnight. The feeling of freedom. My husband wasn’t there at night. I didn’t have to get up, so I was able to sleep easily.”*
(22)

#### 3.6.4. Self-Care for a Lifestyle Involving Family Caregiving

Study participants providing caregiving were engaging in self-care that they were able to perform when leaving the house freely was difficult for them. This included “making efforts to adjust his/her mood despite providing caregiving daily” by going on outings and “alleviating stress through exchanges with visiting nursing care staff” who visited to provide nursing care services.

They “let go of the stress of caregiving through exchanges with friends” when taking classes and the like and engaged in self-care through “exercise in between family caregiving,” such as going bowling and swimming in between their caregiving duties.


*“Sometimes I get frustrated looking after my parents. In times like that, I diffuse aroma therapy oils in the room using a diffuser to calm my mind, or I just smell the oils to adjust my mood.”*
(7)


*“I always tell the staff who come to my house on visits about the situation. Even before they ask, I’ll say things like “it was like this,” “it was like that,” “it was terrible yesterday,” and so on, and talking about it makes me feel better.”*
(22)


*“While my wife is going to rehabilitation, I need to exercise, so I go to the town swimming pool from time to time.”*
(24)

### 3.7. Category Association (Figure 1)

An association chart for the identified categories was created based on the coding family proposed by Glaser. *Role transitions that were predicted and chosen by the participant* was connected to *Release from stressors*, *Sense of fulfillment from new role finding* and *Sense of having extra time*, which were related to *Self-care identified from experiences* and connected to *Positive changes in health awareness and health behaviors associated with role transitions*. Additionally, participants had *Motivation to maintain my health.”*


*“I go to training three days a week, and on the other two days, I walk about 8 kilometers to and from the hospital from home. I’ve been doing this since I retired. I have lower back pain, but I’m training my abdominal and back muscles to be more flexible, so I don’t have any lower back pain these days.”*
(2)

Even if they felt a Sense of fulfillment from new role finding, this actually connected with Temporary fatigue associated with re-employment and/or Temporary sense of fatigue after looking after grandchildren.


*“I feel tired when I get busy with farming.”*
(10)


*“After looking after my grandchildren from morning till night, I’m exhausted when I get home.”*
(18)

The next pattern showed that, even if it was Role transitions that were predicted and chosen by the participant, it was connected to Confusion and sense of isolation felt after retirement. This state was connected to Reluctance to participate in health promoting activities in local communities and correlated with wanting to connect to Hopes to connect with local communities.


*“I’ve been working all the time, so I don’t have any connections in the community. I find it hard to take the first step.”*
(16)

The third pattern was Inevitable role transition to family caregiving connected with states such as Confusion at the role transition to family caregiving and Caregiving lifestyle with no leeway. Further, because Self-care made difficult by family caregiving, this created a Caregiving fatigue that is difficult to cope with by personal effort alone. These situations required the utilization of social resources for caregiving and Self-care for a lifestyle involving family caregiving.


*“Until a year ago, I attended a local sports class twice a week in the evening. I gradually became concerned that I couldn’t leave my mother alone, and I thought I would cause trouble for everyone if I kept missing classes, so I took the plunge and quit sports.”*
(20)

## 4. Discussion

### 4.1. Predicted Role Transition Chosen by the Participant Themselves

First, predicted role transition chosen by the participant themselves appeared to be associated with things such as a sense of achievement in their new role; thus, we discuss the pattern in which role transition went smoothly. In this study, predicted transitions such as “gathering information for post-retirement lifestyle” and “looking after grandchild/ren or accepting family caregiving as planned” seemed to be linked to smoother adaptation to subsequent role transitions.

Meleis’ transition theory indicates that ‘Preparation & knowledge’ is an important factor affecting transition [19]. A past study emphasized that “preparation for change” is needed to enable individuals to utilize the time they have after retirement [20]. Preparing a pre-retirement plan at the workplace connects with a good state of health after retirement [21]. Currently, pre-mandatory retirement education in Japan is being implemented mainly by major corporations and public bodies. However, the number of reports in Japan is insufficient. This study found that, as smooth role transition seemed to be linked to subsequent health, more detailed research on support for the role transition of individuals in their 60s needs to be conducted going forward.

Next, a sense of relaxation appeared to correspond with self-care identified from experience, with participants found to be engaging in exercise during free time and incorporating sleep management into their lifestyles. Retirement is not only an important turning point in a person’s life but also a key source of social stress that can affect a person’s physical and mental health [22,23]. One imported factor that has been reported for good physical function after mandatory retirement is maintaining the physical activities already established by retirees [24]. In Meleis’ transitions theory, self-care behaviors that utilize the temporal and psychological leeway after transition are positioned as indicators of healthy transition (patterns of response/process indicators) [19]. Specifically, practices such as exercise and sleep management, which help individuals adapt to role changes after retirement and rebuild new life rhythms, seemed to assist in restoring balance during the transition. Furthermore, these behaviors, as adaptive actions based on individual agency and experience, correspond to the facilitating conditions for transition emphasized by Meleis [19].

Therefore, the practice of self-care utilizing the free time after retirement can be theoretically explained not merely as an improvement in lifestyle habits, but as part of the process of healthily executing role transitions. Previous research indicates that post-retirement physical activity trajectories are highly individualized, pointing to the necessity of personalized intervention approaches to increase physical activity during the retirement transition period [17,25]. This study showed that, although reduced physical and mental function from the 60s onward was unavoidable, acquired self-care ability was learned over one’s entire lifespan, and it remained on an individual and personal level. These findings indicated that, during the period of role transition in a person’s 60s, creating frameworks to encourage the implementation and continuation of self-care identified from experience was important.

Regarding the *Temporary fatigue associated with re-employment* revealed in this study, Meleis explains that the transition period from retirement to reemployment represents a major change requiring the reconstruction of daily rhythms, self-concept, and social roles, during which physical and psychological fatigue becomes apparent [12]. Furthermore, it is indicated that this fatigue is a temporary phenomenon during the transition period and is expected to diminish as adaptation progresses [26]. To reduce fatigue among re-employed individuals and promote healthy transitions, sufficient rest, self-care support, flexible work arrangements, and social support within the workplace are necessary.

The *Temporary sense of fatigue after looking after grandchildren* demonstrated in this study indicates that while interaction with grandchildren brings psychological fulfillment and social connection, excessive involvement and tension became physical and mental burdens. Previous research has shown that caring for grandchildren can be neither beneficial nor harmful to grandparents’ health [27]. Factors influencing grandparents’ physical and mental health are highly individualized, as the impact varies with family closeness [15,28] and the nature of grandparent-grandchild relationships or living arrangements. Conversely, grandmothers caring for grandchildren for over two years may become more proactive in simple preventive actions (vaccinations, self-examination, etc.) [29]. Based on Meleis’ transitions theory, the transition into the grandparent role is influenced by ‘Preparation & knowledge’ factors [19]. This suggests that when the role is chosen deliberately, it tends to be associated with better health and to contribute to health maintenance, whereas when it is imposed by unavoidable circumstances, the burden increases. To make the transition into the grandparent role a positive change, it is important to respect the grandparents’ freedom to choose their role while providing social support and opportunities for rest.

Despite being able to anticipate mandatory retirement, confusion and sense of isolation felt after retirement were connected to reluctance to participate in health promoting activities in local communities and need to connect with local communities. Meleis’ transition theory indicates that fluctuations in self-concept, such as post-retirement confusion and loneliness, represent typical transitional reactions accompanying “Role loss [9].” Meleis’ transitions theory identifies social support as one factor promoting transition [12]. The extraction of *Hopes to connect with local communities* indicates a “demand for resources necessary to advance transition healthily,” demonstrating consistency with the theory. The emergence of the *Hopes to connect with local communities* can be interpreted as a sign that the individual is beginning to seek out the facilitating factors necessary to achieve a healthy transition. Previous research indicates isolation is a risk factor for fatigue, physical inactivity, and cognitive impairment in elderly people [30], When the death of a spouse, illness, disability, or retirement reduces the size of the persons’ social network, increasing social isolation and loneliness, an early intervention approach to promote social connections is vital [31]. Furthermore, one effect of promoting participation in local communities is not only the prevention of decline in physical and mental functions but the possibility of improving other areas of a person’s happiness through increased social interaction [32]. The Government of Japan is implementing policies to recommend utilizing healthy elderly individuals as human resources to support activities in local communities [2]. To ensure a smooth transition after retirement in their 60s, support needs to be provided to help such individuals easily access information and connect with communities. This support could include providing information regarding regional activities and creating opportunities for local residents to interact, as well as forming local resident organizations. The importance of creating activities such as volunteering, which can support people in starting new activities and providing information, has been demonstrated for the role transition of those in their 60s.

### 4.2. Unavoidable Role Transition to Caregiving

This study found that participants described feeling fatigued in connection with unexpected transitions in caregiving, which seemed to be associated with feelings of fatigue. These points can be said to relate to the importance of preparatory work in role theory as proposed by Meleis’ transitions theory [19]. Past studies have reported that, in the role transition to caregiving, people felt pressure as family caregivers in wondering whether this was their responsibility or choice [32]. Nearly half of caregivers had no option of refusing the role of caregiving and feeling that one does not have a choice correlates with caregiving having harmful effects on health [33]. Investigations of health management for caregivers need to focus on participant backgrounds in role transition to caregiving. A past longitudinal study that investigated the health status of caregivers over 6 years reported a prevalence of 13.6% [34]. The present study also reported insomnia, psychological conflict, and joint pain for chronic sense of fatigue from caregiving. A study conducted in the US reported that caregivers commonly had poor health, including poor sleep and chronic diseases, and that caregivers engaged in negative health behaviors more often than non-caregivers [35].

The present study found that, as self-care throughout the caregiving lifestyle, participants selected methods that they could easily perform by themselves. Therefore, they had discovered these methods themselves as caregiving limited their ability to leave the house. These self-care behaviors can also be interpreted as one of the “facilitators” of transition identified in Meleis’ transitions theory [19]. A past study reported that performing intervention for physical exercise for caregivers greatly reduced their suffering while increasing their level of health, lifestyle quality, sleep quality, physical activity, sense of self-efficacy regarding caregiving and exercise, and preparation for exercise [36]. Both personal factors, such as managing physical activity and diet, as well as external factors, such as medical access and the provision of information, are required to enable self-care for caregivers [37]. Furthermore, it has been pointed out that social participation tends to decline after transitioning to long-term care [38]. In Japan, support for maintaining and improving the health of family caregivers is an important issue. As exercise and health management that can be incorporated into one’s daily caregiving lifestyle are types of self-care that also lead to extended healthy longevity, health support for caregivers needs to be considered by society as a whole.

Finally, as caregivers can rest physically and mentally when caregiving recipients go on outings or overnight stays with nursing care services, we identified a *Utilization of social resources for caregiving* These results can be explained by Meleis’ transitions theory. Meleis pointed out that for a healthy transition, the presence of ‘Community’ and ‘Society’ resources as facilitating conditions is crucial [9]. The need for care service utilization revealed in this study is precisely an expression of caregivers seeking the resources necessary to achieve a healthy transition. Furthermore, ensuring caregivers have adequate rest is positioned as an ‘indicator of healthy transition [12,19]’, as it promotes stability in self-concept, restores daily rhythms, and enhances the sustainability of caregiving roles. Therefore, utilizing social resources that support caregiver rest suggests this is not merely a means of support but an indispensable element for guiding the transition into caregiving roles in a healthy manner. Caregivers need to perform self-care by taking breaks [32], and the lack of time is one factor that affects caregivers’ mental and physical health [39]. In Asian culture, the responsibility to look after elderly members of one’s household is deeply rooted. However, many family caregivers face the problem of society either not noticing that they are providing caregiving or not giving them any support [39]. In Japan, over 20 years have passed since the Long-Term Care Insurance Act was enacted. Nevertheless, family caregivers are still unable to freely adjust their time and take breaks. Therefore, further investigation of utilizing social resources is required going forward.

### 4.3. Suggestions for Nursing Practice

The present study found that role transitions in daily life affected self-care and health status. This supported the finding that health behavior changed during the role-transition period [9]. To specifically target health support for people in their 60s based on the results of this study, the focus on the transition needs to include not only role transition to re-employment but also unavoidable transition to caregiving. Previous studies have pointed out the importance of focusing on the transition to old age, as this transition brings about changes in individuals’ self-perception and role challenges in life [17]. Current policies in Japan are promoting working as much as possible even after mandatory retirement and do not focus on transition to household roles, such as caregiving or looking after grandchildren. Health support for local residents in the transition phase in their 60s requires support for providing information related to local community activities and creating connections with local residents. Furthermore, health support for role transition to caregiving should include support for utilizing social resources and be tailored to individual needs.

### 4.4. Limitations of the Study

This qualitative study did not aim to generalize the results but rather to better understand the phenomena. Individual experience is highly contextual, and even within the similar social environment, the experiences of residents in a geographical location are never the same. We tried to capture a variety of experiences, but we could never do so. Moreover, this study targeted only those in their 60s, without comparisons with other age groups; thus, the conclusion that our findings are characteristic of people in their 60s has limitations. In this study, we enhanced the reliability of our analysis through consensus-building within the research team; however, we did not implement member checking (returning verbatim transcripts or findings) with participants. Consequently, the lack of opportunities to confirm or supplement interpretations from the participants’ own perspectives represents a limitation of this research. However, our findings must be substantial because they provide important suggestions for health support for community-living adults approaching old age. Further validation of the results with more individuals and in different regions is needed to gain further insight.

## 5. Conclusions

The findings suggested that, in predicted role transitions chosen by participants, they tended to experience positive changes in health awareness and health behaviors, although temporary feelings of fatigue were also described in relation to re-employment and grandchild care. Even in anticipated role changes, some participants expressed reluctance to engage in health-promoting activities within the local community.

In cases of inevitable role transition to family caregiving, participants described difficulties in maintaining self-care and feelings of caregiving fatigue that were challenging to manage through personal effort alone. These findings suggest that health support during role transitions in one’s 60s may benefit from including information about community activities and opportunities to build connections with local residents. In addition, support for those transitioning into caregiving roles could focus on facilitating access to social resources and tailoring assistance to individual needs.

To specifically target health support for people in their 60s based on the results of this study, the focus on the transition needs to include not only role transition to re-employment but also unavoidable transition to caregiving.

## Figures and Tables

**Figure 1 healthcare-13-02702-f001:**
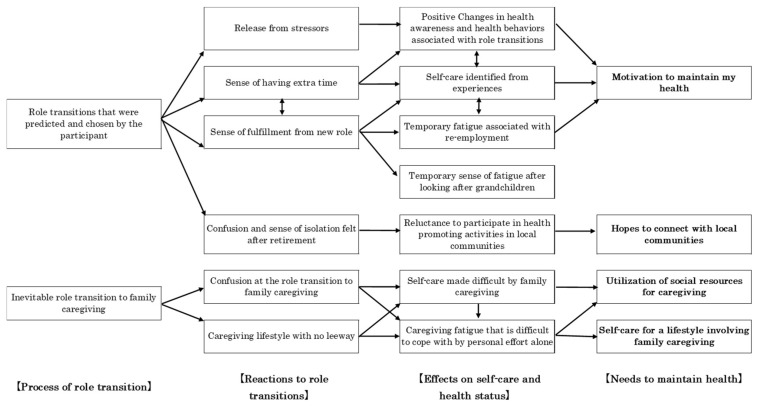
Category Correlations. 【】 indicates the major categoly and □ indicates the category. The direction of the arrows shows the impact and results of the action source and action.

## Data Availability

The data presented in this study are available on request from the corresponding author. The data are not publicly available due to privacy reasons.

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
