# Peer review of "A Qualitative Study of Health-Related Experiences Associated with Lifestyle Role Transitions Among Local Residents in Their 60s"

_healthcare, 2025, doi:10.3390/healthcare13212702_

Round 1

Reviewer 1 Report

Comments and Suggestions for Authors

This is an interesting study examining the influence of role transitions (re-employment, caregiving, grandparenting) on health among older adults in Japan. The topic is very timely and important. I enjoyed reading the paper. I have several suggestions to improve the manuscript further:

1. First, I noticed that role transition is treated monolithically, but caregiving, re-employment, and grandparenting involve different psychological processes and stress mechanisms. The introduction should clarify how these transitions are theoretically distinct.

2. With 26 participants across six focus groups, it will be important to have a discussion of how saturation was determined

3. It will also useful to provide more information on how many coder were involved and how intercoder reliability was ensured. 

4. A recurring issue in the discussion is that much of the narrative simply reiterates what was reported in the results, with minimal theoretical synthesis.  I would like to suggest the authors to relate the findings to broader theoretical frameworks. 

5. One way to strengthen the theoretical framing would be by explicitly addressing why these transitions have varied effects on well-being from aging and life-transition empty next research. Prior work suggests that major life transitions in later adulthood can affect people in very different ways. For some, these changes lead to role loss as leaving work or taking on new family responsibilities can create feelings of identity disruption, reduced purpose, and greater loneliness. For others, the same transitions can bring relief from role strain because stepping away from demanding obligations allows more freedom, increased autonomy, and greater opportunities for self-care and personal goals. This theoretical distinction could help explain the mixed experiences in the current study, where some participants reported improvements in well-being while others described fatigue, isolation, or stress related to caregiving. It may also be helpful to discuss how cultural expectations around family obligations, caregiving, and community involvement in Japan influence how older adults experience and navigate these transitions. See the following relevant theoretical perspective: Cultural contexts differentially shape parents’ loneliness and wellbeing during the empty nest period. (2024). Communications Psychology, 2(1), 105.

6. While quotes are provided, there is little reporting on pattern prevalence. Will be useful to elaborate on which experiences were common versus rare?

Author Response

Response to reviewer 1

  1. First, I noticed that role transition is treated monolithically, but caregiving, re-employment, and grandparenting involve different psychological processes and stress mechanisms. The introduction should clarify how these transitions are theoretically distinct.

Reply:

Thank you for your comments.

In Meleis' transition theory, caregiving and re-employment were classified as situational transitions, while becoming a grandparent was categorized as a developmental role transition. I added this point to the introduction. For this revision, I considered multiple theories including role theory, activity theory, and continuity theory. However, I concluded that using various theories could potentially cause confusion. Therefore, I ultimately revised the analysis to utilize Meleis’ transition theory. Please refer to the Introduction and Discussion sections.

  1. With 26 participants across six focus groups, it will be important to have a discussion of how saturation was determined.

Reply

Thank you for your feedback. We have added details regarding the determination of saturation. Please refer to the Figure 1.

  1. It will also useful to provide more information on how many coder were involved and how intercoder reliability was ensured.

Reply

Thank you for your feedback. The number of codes is 110. To ensure the reliability of our analysis, we made the following additions.

Please refer to Result   3.2.     Interview Results.

  1. A recurring issue in the discussion is that much of the narrative simply reiterates what was reported in the results, with minimal theoretical synthesis. I would like to suggest the authors to relate the findings to broader theoretical frameworks.

Reply

In this revision, we considered incorporating multiple theories,
but ultimately determined that Meleis’ theory of transition best fits the findings of this study. Accordingly, we substantially revised the discussion. Please refer to the Discussion sections.

  1. One way to strengthen the theoretical framing would be by explicitly addressing why these transitions have varied effects on well-being from aging and life-transition empty next research. Prior work suggests that major life transitions in later adulthood can affect people in very different ways. For some, these changes lead to role loss as leaving work or taking on new family responsibilities can create feelings of identity disruption, reduced purpose, and greater loneliness. For others, the same transitions can bring relief from role strain because stepping away from demanding obligations allows more freedom, increased autonomy, and greater opportunities for self-care and personal goals. This theoretical distinction could help explain the mixed experiences in the current study, where some participants reported improvements in well-being while others described fatigue, isolation, or stress related to caregiving. It may also be helpful to discuss how cultural expectations around family obligations, caregiving, and community involvement in Japan influence how older adults experience and navigate these transitions. See the following relevant theoretical perspective: Cultural contexts differentially shape parents’ loneliness and wellbeing during the empty nest period. (2024). Communications Psychology, 2(1), 105.

Reply

Thank you for providing the reference paper.

In this revision, we deepened our theoretical examination based on Meleis’ Transition Theory as the foundational theory supporting the positive health outcomes following role transition. While we incorporated elements of Continuity Theory in the Introduction, the primary theoretical framework for the Discussion section is Transition Theory. Please refer to the Discussion section.

  1. While quotes are provided, there is little reporting on pattern prevalence. Will be useful to elaborate on which experiences were common versus rare?

Reply
Thank you for your feedback.

We reviewed major papers detailing the frequency of re-employment, caregiving, and grandchild care. However, due to the ambiguous definitions of transition across papers and the diverse data sources, we found it difficult to determine a consistent rate. Nevertheless, we have added the following paper and revised the introduction. Please refer to the Introduction section.

Added references

  1. Zanasi, F.; Arpino, B.; Bordone, V.; Hank, K. The prevalence of grandparental childcare in Europe: a research update. European journal of ageing 2023, 20, 37, doi:10.1007/s10433-023-00785-8.
  2. Maestas, N. Back to Work: Expectations and Realizations of Work after Retirement. J Hum Resour 2010, 45, 718-748, doi:10.1353/jhr.2010.0011.
  3. Roth, D.L.; Haley, W.E.; David Rhodes, J.; Sheehan, O.C.; Huang, J.; Blinka, M.D.; Yuan, Y.; Irvin, M.R.; Jenny, N.; Durda, P.; et al. Transitions to family caregiving: enrolling incident caregivers and matched non-caregiving controls from a population-based study. Aging Clin Exp Res 2020, 32, 1829-1838, doi:10.1007/s40520-019-01370-9.

Reviewer 2 Report

Comments and Suggestions for Authors
  1. Main issue addressed by research

The study seeks to clarify how social role transitions in the 60-year-old age group (re-employment, caring for grandchildren, caring for family members) relate to health perception, to inform public health measures linked to everyday life. The central issue is the association between role change in active ageing and perceived well-being/health. This is a very important study for the entire Western world and developing countries that are facing, and will face, the problem of ageing in their societies.

  1. Originality and relevance

The topic is socially relevant: active ageing and extending working life are international priorities (WHO, “Decade of Healthy Ageing”), as well as local ones (in the Japanese context, with its very pronounced population ageing).

There is a specific gap: little is known about how people experience multiple and simultaneous role transitions (retirement, re-employment, caring for grandchildren, family caregiving) and how this affects their health.

However, the scientific originality of this study is very good: there are already studies on active ageing, caregiving and transitions in retirement, but this work combines the three roles in the same qualitative analysis and in a community context, which, in my opinion, gives it substantial novelty.

  1. Contribution in relation to other published works

It adds an integrated perspective: it shows that transitions can be anticipated (e.g., re-employment, caring for grandchildren) or inevitable (e.g., caregiving), and that this difference influences the impact on individuals' perceived health.

This study identifies important contributions from countries' health and social policies that stem from community support needs and public policies tailored to the profile of older people in transition.

It seems to me that the study reinforces the concept of social roles as social determinants of health and highlights the importance of socialization and self-care resources, in comparison with existing literature.

However, the authors could further the discussion with more international studies on role transitions and health in older adults, to broaden and deepen the comparison and theoretical advancement on the topic.

  1. Methodological improvements to consider

The title of the paper should indicate the study methodology.

The keywords are not in accordance with MeSH descriptors.

The authors should clarify and correct the information on the number of participants in the study: in the abstract, they indicate 23 participants (line 16), and in the methods/results section, on line 145 and in Table 1, they indicate 26 residents.

Ethical issues are safeguarded and correct.

The recruitment process should be better justified: convenience sampling via municipal employees may introduce bias. It is also necessary to discuss the limitations of this methodological option.

Regarding focus groups, there is data that should be completed and added: details are missing on duration, question script, exact location, presence of non-participants, theoretical saturation, and number of coders.

Regarding the team that interviewed the residents, the authors should provide information on who conducted the interviews, their training, and any biases.

The authors describe “qualitative descriptive analysis,” but refer to Glaser’s “coding family.” This methodological decision needs to be clarified and aligned.

About data triangulation, the authors do not indicate whether there was member checking or whether transcripts or results were returned to participants. It is recommended that this absence be at least acknowledged as a limitation of the study.

Regarding the characterization of participants, the authors could add relevant sociodemographic information, which in my opinion is lacking, such as education level, marital status, length of caregiving, and family composition.

In the discussion section, it is suggested that the discussion be reinforced with comparative international literature to strengthen the overall importance of the paper and the study.

  1. Conclusions versus scientific evidence

The conclusions are consistent with the data.

The results illustrate that re-employment and caring for grandchildren bring benefits, but also fatigue; caregiving implies overload and less self-care and eventually mental health issues.

However, statements such as participants experiencing better overall health than before retirement go beyond subjective perceptions without the support of objective measures. Here there is overinterpretation that should be cautioned against.

The authors should rephrase terms such as ‘participants reported feeling...’ and acknowledge that these are self-reported perceptions.

  1. References

Most are appropriate (WHO, Japanese national reports, Meleis' theory) and ensure scientific relevance. However, the authors should add more recent international sources on role transitions and ageing outside Japan.

  1. Additional comments on tables and figures

Table 1: presents ID, gender, age and role, but it is not very informative. It is recommended to add education, marital status, occupation before/after retirement, and hours spent on caregiving.

Table 2: lists categories and subcategories, but the formatting makes it difficult to read. Reorganize into a clear hierarchy.

Figure 1: improve the aesthetics of the arrows. The explanatory caption needs to be improved to better articulate with Table 2.

Final summary of the opinion

The article is highly relevant and has the potential to contribute to the field of public health and community nursing in active ageing, but it requires further revision. Improvements should focus on correcting the title and keywords, methodological consistency, detailed description of qualitative procedures (including a complete report according to the 32 COREQ items), standardization of N, and improving the clarity of tables/figures.

Author Response

Response to reviewer 2

1.Main issue addressed by research

The study seeks to clarify how social role transitions in the 60-year-old age group (re-employment, caring for grandchildren, caring for family members) relate to health perception, to inform public health measures linked to everyday life. The central issue is the association between role change in active ageing and perceived well-being/health. This is a very important study for the entire Western world and developing countries that are facing, and will face, the problem of ageing in their societies.

Reply

Thank you for your comments.

2.Originality and relevance

The topic is socially relevant: active ageing and extending working life are international priorities (WHO, “Decade of Healthy Ageing”), as well as local ones (in the Japanese context, with its very pronounced population ageing).

There is a specific gap: little is known about how people experience multiple and simultaneous role transitions (retirement, re-employment, caring for grandchildren, family caregiving) and how this affects their health.

However, the scientific originality of this study is very good: there are already studies on active ageing, caregiving and transitions in retirement, but this work combines the three roles in the same qualitative analysis and in a community context, which, in my opinion, gives it substantial novelty.

Reply

Thank you for your comments.

3.Contribution in relation to other published works

It adds an integrated perspective: it shows that transitions can be anticipated (e.g., re-employment, caring for grandchildren) or inevitable (e.g., caregiving), and that this difference influences the impact on individuals' perceived health.

This study identifies important contributions from countries' health and social policies that stem from community support needs and public policies tailored to the profile of older people in transition.

It seems to me that the study reinforces the concept of social roles as social determinants of health and highlights the importance of socialization and self-care resources, in comparison with existing literature.

However, the authors could further the discussion with more international studies on role transitions and health in older adults, to broaden and deepen the comparison and theoretical advancement on the topic.

Reply

Thank you for your advice.

In this revision, we further utilized Meleis’ role transition to provide a theoretical framework. Please refer to the discussion section. We have added the following 15 references to the introduction and discussion sections.

  1. Zanasi, F.; Arpino, B.; Bordone, V.; Hank, K. The prevalence of grandparental childcare in Europe: a research update. European journal of ageing 2023, 20, 37, doi:10.1007/s10433-023-00785-8.
  2. Maestas, N. Back to Work: Expectations and Realizations of Work after Retirement. J Hum Resour 2010, 45, 718-748, doi:10.1353/jhr.2010.0011.
  3. Roth, D.L.; Haley, W.E.; David Rhodes, J.; Sheehan, O.C.; Huang, J.; Blinka, M.D.; Yuan, Y.; Irvin, M.R.; Jenny, N.; Durda, P.; et al. Transitions to family caregiving: enrolling incident caregivers and matched non-caregiving controls from a population-based study. Aging Clin Exp Res 2020, 32, 1829-1838, doi:10.1007/s40520-019-01370-9.
  4. Atchley, R.C. A continuity theory of normal aging. The Gerontologist 1989, 29, 183-190, doi:10.1093/geront/29.2.183.
  5. Meleis, A.I.; Sawyer, L.M.; Im, E.O.; Hilfinger Messias, D.K.; Schumacher, K. Experiencing transitions: an emerging middle-range theory. ANS Adv Nurs Sci 2000, 23, 12-28, doi:10.1097/00012272-200009000-00006.
  6. Haley, W.E.; Roth, D.L.; Sheehan, O.C.; Rhodes, J.D.; Huang, J.; Blinka, M.D.; Howard, V.J. Effects of Transitions to Family Caregiving on Well-Being: A Longitudinal Population-Based Study. Journal of the American Geriatrics Society 2020, 68, 2839-2846, doi:10.1111/jgs.16778.
  7. Di Gessa, G.; Glaser, K.; Tinker, A. The impact of caring for grandchildren on the health of grandparents in Europe: A lifecourse approach. Social science & medicine (1982) 2016, 152, 166-175, doi:10.1016/j.socscimed.2016.01.041.
  8. Danielsbacka, M.; Křenková, L.; Tanskanen, A.O. Grandparenting, health, and well-being: a systematic literature review. European journal of ageing 2022, 19, 341-368, doi:10.1007/s10433-021-00674-y.
  9. Zhu, J.; Xu, L.; Sun, L.; Qin, D. Negative life events, sleep quality, and depression among older adults in Shandong Province, China: A conditional process analysis based on economic income. Geriatrics & gerontology international 2024, 24, 751-757, doi:10.1111/ggi.14914.
  10. Morgan, G.S.; Willmott, M.; Ben-Shlomo, Y.; Haase, A.M.; Campbell, R.M. A life fulfilled: positively influencing physical activity in older adults - a systematic review and meta-ethnography. BMC Public Health 2019, 19, 362, doi:10.1186/s12889-019-6624-5.
  11. McDonald, S.; O'Brien, N.; White, M.; Sniehotta, F.F. Changes in physical activity during the retirement transition: a theory-based, qualitative interview study. Int J Behav Nutr Phys Act 2015, 12, 25, doi:10.1186/s12966-015-0186-4.
  12. Schumacher, K.L.; Meleis, A.I. Transitions: a central concept in nursing. Image J Nurs Sch 1994, 26, 119-127, doi:10.1111/j.1547-5069.1994.tb00929.x.
  13. Baker, L.A.; Silverstein, M. Preventive health behaviors among grandmothers raising grandchildren. The journals of gerontology. Series B, Psychological sciences and social sciences 2008, 63, S304-311, doi:10.1093/geronb/63.5.s304.
  14. Wang, X.R.; Liu, S.X.; Robinson, K.M.; Shawler, C.; Zhou, L. The impact of dementia caregiving on self-care management of caregivers and facilitators: a qualitative study; 1346-3500; Jan 2019; pp. 23-31.
  15. Li, L.; Wister, A.V.; Lee, Y.; Mitchell, B. Transition Into the Caregiver Role Among Older Adults: A Study of Social Participation and Social Support Based on the Canadian Longitudinal Study on Aging. The journals of gerontology. Series B, Psychological sciences and social sciences 2023, 78, 1423-1434, doi:10.1093/geronb/gbad075.

4.Methodological improvements to consider

4-1)The title of the paper should indicate the study methodology.

The keywords are not in accordance with MeSH descriptors.

The authors should clarify and correct the information on the number of participants in the study: in the abstract, they indicate 23 participants (line 16), and in the methods/results section, on line 145 and in Table 1, they indicate 26 residents.

Ethical issues are safeguarded and correct.

Reply

◆Thank you for pointing that out. I have changed the title.

Before correction: The impact of role transition on health in local residents in their 60s

After correction: A Qualitative Study on the impact of role transition on health in local residents in their 60s

◆We have modified the keywords to conform to the MeSH descriptors.

Before correction Keywords: role transition, re-employment, looking after grandchildren, caring for family

After correction Keywords: Role; Adaptation, Psychological; Health Status; Quality of Life; Aged

◆The abstract contained an error, so we are correcting the number of people in the abstract.

Before correction  Abstract:

Methods: We conducted focus group interviews with 23 residents and analyzed them qualitatively and inductively.

After correction Methods: We conducted focus group interviews with 26 residents and analyzed them qualitatively and inductively.

4-2)The recruitment process should be better justified: convenience sampling via municipal employees may introduce bias. It is also necessary to discuss the limitations of this methodological option.

Reply

This study employed convenience sampling, as research participants were recruited through municipal employees. While this method offers practical advantages for study implementation—such as ease of contact with participants and ethical considerations—it carries the potential bias of favoring residents connected to the municipality and may not represent the entire local population.
Therefore, we added a limitation stating that the study's findings require careful interpretation when generalizing to the entire elderly population of the region.

Please refer to the Limitation section.

4-3)Regarding focus groups, there is data that should be completed and added:

details are missing on duration, question script, exact location, presence of non-participants, theoretical saturation, and number of coders.

Reply

Thank you for your advice. I apologize for not having included the number of extracted codes.

The number of codes is 110. This was confirmed through discussion and consensus among researchers, with careful consideration given to reflecting data diversity while avoiding arbitrary subdivision. As a result, they were consolidated into 54 subcategories and 20 categories, striving to ensure the reliability and validity of the analysis. This point has been added to the analysis methodology.

Please refer to Result   3.2.     Interview Results.

The interview script has been added as Supplementary Material 1: Interview guides. Please review it.

4-4

(1) Regarding the team that interviewed the residents, the authors should provide information on who conducted the interviews, their training, and any biases.

Reply
The first author conducted the interviews, and it was done after practice.

Please refer to 2. Materials and Methods2.4. Data Collection and Analysis Methods.

(2) The authors describe “qualitative descriptive analysis,” but refer to Glaser’s “coding family.” This methodological decision needs to be clarified and aligned.

Reply
Thank you for your feedback. We have added the basis for using Glaser's “coding family.” Please refer to 2. Materials and Methods2.4. Data Collection and Analysis Methods.

(3) About data triangulation, the authors do not indicate whether there was member checking or whether transcripts or results were returned to participants. It is recommended that this absence be at least acknowledged as a limitation of the study.

Reply

In this study, to enhance the validity of the analysis, we engaged in repeated discussions with multiple researchers specializing in public health nursing, cross-referencing verbatim transcripts and categories to reach consensus. However, we did not conduct member checking with participants (the procedure of returning verbatim transcripts or findings for their confirmation). Therefore, we recognize that the lack of verification or supplementation of interpretations by the participants themselves constitutes a methodological limitation of this study, and we have added this limitation accordingly. Please refer to 2. Materials and Methods2.4. Data Collection and Analysis Methods and 4. Discussion 4.4.            Limitations of the study.

(4) Regarding the characterization of participants, the authors could add relevant sociodemographic information, which in my opinion is lacking, such as education level, marital status, length of caregiving, and family composition.

Reply

Family composition, care recipient level, and pre-care occupation have been added (Table 2). Education level and duration of caregiving are not available. We apologize for this. Please refer to Table 2.

(5)In the discussion section, it is suggested that the discussion be reinforced with comparative international literature to strengthen the overall importance of the paper and the study.

Reply
Thank you for your advice.

With this revision, We have added the following paper and revised the discussion.

  1. Conclusions versus scientific evidence

The conclusions are consistent with the data.

5-1) The results illustrate that re-employment and caring for grandchildren bring benefits, but also fatigue; caregiving implies overload and less self-care and eventually mental health issues.

Reply

Thank you for your advice.

5-2) However, statements such as participants experiencing better overall health than before retirement go beyond subjective perceptions without the support of objective measures. Here there is overinterpretation that should be cautioned against.

Reply
Thank you for your feedback. We have changed the category name to “Health Promotion Through Role Transition.”

5-3) The authors should rephrase terms such as ‘participants reported feeling...’ and acknowledge that these are self-reported perceptions.

Reply
Thank you for your advice. We have checked that there is no term such as ‘participants reported feeling...’. present in the main text.

6. References

Most are appropriate (WHO, Japanese national reports, Meleis' theory) and ensure scientific relevance. However, the authors should add more recent international sources on role transitions and ageing outside Japan.

Reply
Thank you for your feedback. We have added the following 15 references to the introduction and discussion sections.

7.Additional comments on tables and figures

Table 1: presents ID, gender, age and role, but it is not very informative. It is recommended to add education, marital status, occupation before/after retirement, and hours spent on caregiving.

Table 2: lists categories and subcategories, but the formatting makes it difficult to read. Reorganize into a clear hierarchy.

Figure 1: improve the aesthetics of the arrows. The explanatory caption needs to be improved to better articulate with Table 2.

Reply

Thank you for your advice.

Table 1 now includes information on occupation before and after retirement and family composition. While caregiving hours are unknown, we have added the caregiving level. Please note that we do not have information on educational background.

Table 2 has increased line spacing to improve readability.

Figure 1 now has colored arrows and a revised caption.

Final summary of the opinion

The article is highly relevant and has the potential to contribute to the field of public health and community nursing in active ageing, but it requires further revision. Improvements should focus on correcting the title and keywords, methodological consistency, detailed description of qualitative procedures (including a complete report according to the 32 COREQ items), standardization of N, and improving the clarity of tables/figures.

Reply
Thank you for your advice.

Round 2

Reviewer 1 Report

Comments and Suggestions for Authors

I appreciate the substantial revisions made to the manuscript and the authors’ responsiveness to earlier concerns. The manuscript is clearer in its theoretical framing and provides a much stronger discussion. That said, I see a few areas where additional clarification would further strengthen the paper:

1. The Methods now indicate that transcripts were coded collaboratively and that consensus was reached through repeated discussion among researchers. The description would benefit from greater transparency. For instance, how many coders were involved in the initial coding? Were transcripts coded independently before reconciliation? If possible, please clarify whether any formal intercoder reliability indices (e.g., Cohen’s kappa) were calculated or why such indices were not deemed appropriate in this context.

2. I noticed that some sections still employ causal-sounding language that may overstate what can be inferred from a qualitative descriptive design. I recommend softening such phrasing throughout the abstract, discussion, and conclusion. Expressions such as “participants described,” “the findings suggest,” or “these experiences appeared to be associated with” would better reflect the interpretive nature of the work and avoid over-interpretation. Similarly, I would encourage the authors to revisit the manuscript title, which currently has a somewhat causal tone. A more descriptive phrasing might better reflect the qualitative design and avoid over-interpretation.

Reviewer 2 Report

Comments and Suggestions for Authors

Thank you very much for your corrections to improve the paper.

Author Response

Reviewer 2

  • Comments and Suggestions for Authors

Thank you very much for your corrections to improve the paper.

Submission Date

18 August 2025

Date of this review

04 Oct 2025 18:09:49

  • Reply

Thank you for taking the time to review my work despite your busy schedule.

We are truly grateful.

Round 3

Reviewer 1 Report

Comments and Suggestions for Authors

The authors have addressed my comments well. I appreciate all their efforts. The paper is ready for publication.